# Activation of neuronal adenosine A₁ receptors causes hypothermia through central and peripheral mechanisms

Haley S. Province[1]☯, Cuiying Xiao[1]☯, Allison S. Mogul[1], Ankita Sahoo[2], Kenneth A. Jacobson [3], Ramón A. Piñol[1], Oksana Gavrilova[2], Marc L. Reitman [1]*

1 Diabetes, Endocrinology, and Obesity Branch, National Institute of Diabetes and Digestive and Kidney Diseases, NIH, Bethesda, Maryland, United States of America, 2 Mouse Metabolism Core, National Institute of Diabetes and Digestive and Kidney Diseases, NIH, Bethesda, Maryland, United States of America, 3 Laboratory of Bioorganic Chemistry, National Institute of Diabetes and Digestive and Kidney Diseases, NIH, Bethesda, Maryland, United States of America

☯ These authors contributed equally to this work.
* marc.reitman@nih.gov

**Data Availability Statement:** All relevant data are within the paper and its Supporting Information file.

## Abstract

Extracellular adenosine, a danger signal, can cause hypothermia. We generated mice lacking neuronal adenosine A₁ receptors (A₁AR, encoded by the *Adora1* gene) to examine the contribution of these receptors to hypothermia. Intracerebroventricular injection of the selective A₁AR agonist (Cl-ENBA, 5'-chloro-5'-deoxy-$N^6$-endo-norbornyladenosine) produced hypothermia, which was reduced in mice with deletion of A₁AR in neurons. A non-brain penetrant A₁AR agonist [SPA, $N^6$-(p-sulfophenyl) adenosine] also caused hypothermia, in wild type but not mice lacking neuronal A₁AR, suggesting that peripheral neuronal A₁AR can also cause hypothermia. Mice expressing Cre recombinase from the *Adora1* locus were generated to investigate the role of specific cell populations in body temperature regulation. Chemogenetic activation of Adora1-Cre-expressing cells in the preoptic area did not change body temperature. In contrast, activation of Adora1-Cre-expressing dorsomedial hypothalamus cells increased core body temperature, concordant with agonism at the endogenous inhibitory A₁AR causing hypothermia. These results suggest that A₁AR agonism causes hypothermia via two distinct mechanisms: brain neuronal A₁AR and A₁AR on neurons outside the blood-brain barrier. The variety of mechanisms that adenosine can use to induce hypothermia underscores the importance of hypothermia in the mouse response to major metabolic stress or injury.

## Introduction

Extracellular adenosine, a product of excessive ATP breakdown such as from extreme metabolic demand or tissue injury, has been called a 'retaliatory metabolite' or danger signal [1]. Due to rapid (i.e., seconds) cellular uptake and metabolism, extracellular adenosine levels are typically low [2] and many of its effects occur locally. Adenosine elicits protective physiology to diminish metabolic demand, reduce injury, and orchestrate recovery from the instigating

**Funding:** This research was supported in part by the Intramural Research Program of the National Institutes of Health, National Institute of Diabetes and Digestive and Kidney Diseases (ZIA DK075062 and ZIA DK075063 to MLR and ZIA DK070002 to OG). The funders had no role in study design, data collection and analysis, decision to publish, or preparation of the manuscript.

**Competing interests:** The authors have declared that no competing interests exist.

'extreme' physiology [3]. The adenosine A$_1$ receptor (A$_1$AR) is one of the four adenosine G protein-coupled receptors and is primarily coupled to G$_{i/o}$ [4]. Its structure has been solved [5]. A$_1$AR is very widely expressed and agonism at this receptor has cardiovascular (bradycardia, protection from ischemia-reperfusion injury), nervous system (antinociception, anti-seizure, protection from ischemia injury, sleep), anti-inflammatory, antidiuretic, and tissue protective (lung, kidney) actions, among other effects [6].

Clinically, hypothermia is used to reduce injury after neonatal hypoxia or cardiopulmonary resuscitation [7]. The hypothermia is typically produced by physical cooling. However, a wide variety of pharmacologic agents can cause hypothermia (see [8]), including adenosine [9], and there is interest in using such agents clinically. This can be readily investigated using mice, as their small size makes them a sensitive, responsive model for exploring mammalian thermal physiology [10]. In the brain, the preoptic area (POA) plays a major integratory role in the control of body temperature [11]. For example, this region has neurons that can drive hypothermia/torpor [12], so this brain region is a prime candidate for pharmacologic actions.

A number of lines of evidence indicate a role for A$_1$AR in regulation of core body temperature (Tb), particularly hypothermia. Classic pharmacologic studies identified agonism at brain A$_1$AR as able to cause hypothermia [13] and there is interest in using A$_1$AR agonists for therapeutic hypothermia [14]. Selective agonist microinjections and other localization studies suggested a role in hypothermia for A$_1$AR in multiple brain regions, including the POA [15–18]. However, we and others have determined that agonists selective for each of the four adenosine receptors can cause hypothermia [19–22]. Also, while a role for A$_1$AR in torpor has been proposed [23, 24], neither A$_1$AR nor any of the other ARs is required for torpor [21, 25].

Understanding the role of A$_1$AR in temperature regulation is complicated by the very wide distribution of the A$_1$AR throughout the brain [6, 26] and its presence in both neurons and glia [27]. The highest brain A$_1$AR concentrations are in portions of the hippocampus, thalamus, and cerebral and cerebellar cortex. Interestingly, A$_1$AR levels are much lower in the hypothalamus, including the POA [26]. Here we investigated where A$_1$AR agonists might act to cause hypothermia.

## Materials and methods

### Mice

Mice at 3–12 months of age were housed at 21–22°C with a 12:12-h dark:light cycle (lights on at 0600) in a clean, conventional facility with paper bedding (7099-TEK-fresh, Envigo, Indianapolis, IN) and *ad libitum* access to water and chow (7022, 15% kcal fat, Envigo Inc, Madison, WI). Experiments were approved by the NIDDK Institutional Animal Care and Use Committee (protocol K016-DEOB-20). Mice were studied >7 days after any operation or prior treatment, and singly housed after telemeter implantation. Reuse of mice tends to reduce physical activity levels during drug testing, presumably due to acclimatization to handling. No specific effort was made to acclimatize mice in individual experiments. Mice were euthanized by exposure to carbon dioxide followed by cervical dislocation.

Male C57BL/6J mice were purchased from Jackson Laboratories, Bar Harbor, ME (JAX; #000664). Ai6 mice carrying Cre-dependent fluorescent protein ZsGreen [28] were purchased from JAX (#007906). Male *Adora1$^{-/-}$* mice on a C57BL/6J background were provided by Dr. Jurgen Schnermann, NIDDK [29] and genotyped as described [21]. *Adora1$^{fl/fl}$* mice [30] were generously provided by Dr. Robert Greene, UTSW, and genotyped by PCR. Primers x692 (forward, 5'-CCACCATTATCTGGCTCCCAT) and x691 (reverse, 5'-GCTGAGTCAC-CACTGTCTTGT) produce 268-bp (wild-type allele) and 302-bp (flox allele) products. Primers x692 and x708 (reverse, 5'-GCTCCCCTGTCTGACTGAAG) produce a ~230-bp product from

a deleted flox allele. Syn1-Cre mice expressing Cre in neurons [31] were purchased from JAX (#003966) and genotyped (~300 bp product) using primers x682 (forward, 5'-CTCAGCGC TGCCTCAGTCT) and x683 (reverse, 5'-GCATCGACCGGTAATGCA). Offspring (both sexes) of *Adora1*$^{fl/fl}$ x *Adora1*$^{fl/+}$;Syn1-Cre/+ and *Adora1*$^{del/fl}$ x *Adora1*$^{fl/+}$;Syn1-Cre/+ matings were used for neuronal deletion experiments.

## Generation of Adora1-Cre mice

The Adora1-Cre mouse was made in a B6D2F1/J founder at the NHLBI Transgenic Core using CRISPR/Cas9 to insert a T2A peptide sequence and Cre recombinase at the *Adora1* stop codon. The T2A sequence adds EGRGSLLTCGDVEENPG to the C terminus of A$_1$AR and a proline to the N terminus of Cre. It is expected that the two proteins are translated in a 1:1 ratio [32] and that the added T2A sequence may reduce A$_1$AR function [33, 34]. The founder was bred to C57BL/6J once. Presence of Adora1-Cre (553 bp) vs wild type (318 bp) alleles was determined by PCR using primers x617 (common forward, 5'-AAGTTCCGGGTCACCTTTC T), cre1 (Cre reverse, 5'-CCTGTTTTGCACGTTCACCG), and x630 (*Adora1* reverse, 5'- AC TCCAAACCTCCTCCAGGT). While homozygous Adora1-Cre mice are fertile and appear grossly normal, only heterozygous Adora1-Cre mice were used in this study. The Adora1-Cre mouse is available from the corresponding author.

## Compounds

Compounds (source; vehicle) were obtained as follows: Cl-ENBA [(±)-5'-chloro-5'-deoxy-*N*$^6$-endo-norbornyladenosine, Tocris 3576; 10% DMSO], SPA [*N*$^6$-(p-sulfophenyl)adenosine, Sigma S198; saline], CNO (clozapine-N-oxide, Sigma C0832; saline).

## Quantitative PCR and RT-PCR

DNA and RNA were extracted (Allprep DNA/RNA micro Kit, Qiagen, Germantown, MD) from whole brain, excluding cerebellum and brain stem. RNA was reverse transcribed (Transcriptor High Fidelity cDNA Synthesis Kit, Roche, Indianapolis, IN). DNA and cDNA were quantified (QuantStudio 7 Flex Real-Time PCR System, Applied Biosystems, Waltham, MA) using SYBR green. *Adora1* mRNA primers were x711 (forward, 5'-CATTGGGCCACAGACC TACT) and x713 (reverse, 5'-TGTACCGGAGAGGGATCTTG), normalized to 18S RNA using primers x530 (forward, 5'-AGTCCCTGCCCTTTGTACACA), and x531 (reverse, 5'-CGATCC GAGGGCCTCACTA).

## Telemetric monitoring of body temperature and activity

Core body temperature (Tb) and physical activity were continuously measured by telemetry, using G2 E-Mitters implanted intraperitoneally, ER4000 energizer/receivers, and VitalView software (v 5.0 Starr Life Sciences, Oakmont, PA), with data collected each minute [35].

## Surgical procedures

Mice were anesthetized with ketamine/xylazine (80/10 mg/kg, i.p.) and placed in a stereotaxic instrument (Digital Just for Mouse Stereotaxic Instrument, Stoelting). Ophthalmic ointment (Puralube, Dechra) was applied. Post-surgery mice received subcutaneous sterile saline injections to prevent dehydration and an analgesic (buprenorphine; 0.1 mg/kg, i.p.). G2 E-Mitters were implanted intraperitoneally as described [35].

## Central infusions

Sterile guide cannulas (5.25 mm, 26 gauge; Plastics One, Roanoke, VA) were unilaterally implanted into the lateral ventricle (coordinates relative to bregma: -0.34 mm anterior, 1.0 mm lateral, 1.7 mm ventral) and fixed with dental cement (Parkell, Edgewood, NY). Compounds in 5 μl were infused (0.5 μl/min) through a 33-gauge cannula protruding 0.5 mm past the tip of the guide cannula using PE-50 tubing fitted to a 5 μl syringe (Hamilton, Reno, NV) on a dual syringe pump (KD Scientific, Holliston, MA).

## Virus injections

All injections (200 nl) were done with pulled-glass pipettes (pulled 20–40 μm tip diameter; 0.275 ID, 1 mm OD, Wilmad Lab Glass) at a visually controlled rate of 50 nl per min with an air pressure system regulator (Grass Technologies, Model S48 Stimulator). The pipette was kept in place for 5 min after injection. The viruses AAV8-hSyn-DIO-hM3Dq-mCherry (gift from B. Roth; Addgene viral prep # 44361-AAV8 [36]) and AAV8-hSyn-DIO-hM4Di-mCherry (gift from B. Roth; Addgene viral prep # 44362-AAV8 [36]) were injected bilaterally into the preoptic area (POA, coordinates relative to bregma: 0.35 mm anterior, ±0.3 mm lateral, -5.25 mm ventral) or dorsomedial hypothalamus (DMH, coordinates relative to bregma: 1.85 mm posterior, ±0.25 mm lateral, -5.2 mm ventral) of Adora1-Cre mice. This batch of AAV8-hSyn-DIO-hM4Di-mCherry was functional in other experiments in our lab that were not part of this project. The hM4Di-mCherry is a fusion protein, so if mCherry is expressed, then hM4Di is also expressed.

## Chemogenetics experimental procedure

hM3Dq and hM4Di were activated by CNO (1 mg/kg, i.p.) or saline vehicle. After completion of all experiments, mice were anesthetized (chloral hydrate, 500 mg/kg, i.p.), perfused transcardially with 0.9% saline followed by 10% neutral buffered formalin, the brain was removed, and reporter expression was visualized by immunohistochemistry [33]. Mice without hM3Dq or hM4Di expression in the target area on at least one side were excluded from analysis.

## Experimental design and statistical analysis

Drug treatments were typically administered in crossover design using randomized treatment order. Hypothermia was assessed as the mean Tb from 0 to 60 minutes after dosing. Hyperthermia was assessed from 60 to 120 minutes after dosing to exclude the handling-induced Tb increase. Physical activity was assessed as the mean from 10 to 60 (hypothermia experiments to avoid the first 10 min after handling) or 60 to 120 (hyperthermia experiments) minutes after dosing. In crossover experiments, paired *t*-tests were used to compare the within mouse effect of drug vs vehicle. Unpaired *t*-tests were used to compare the within mouse drug vs vehicle effect between genotypes. Statistical significance was defined as 2-tailed P <0.05. Data are presented as mean ±SEM. No statistical methods were used to pre-determine sample size. Data, keyed to each figure, are available in the S1 Data.

## Image capture and processing

Images were captured using an Olympus BX61 motorized microscope with Olympus BX-UCB hardware (VS120 slide scanner) and processed (including contrast adjustments) using OlyVIA software (Olympus).

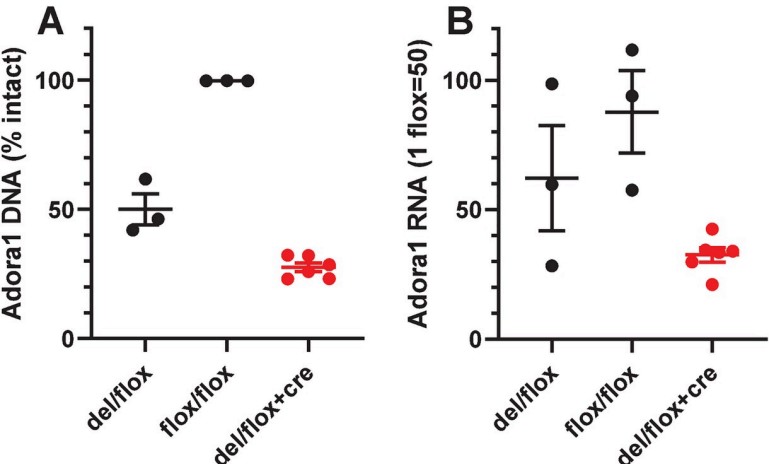

**Fig 1. Levels of *Adora1* DNA and RNA in mouse brain.** (A) *Adora1* DNA levels in brains from control (black, n = 3 del/flox and n = 3 flox/flox) and Syn1-Cre (red, n = 6 del/flox;Syn1-Cre) mice. The % intact is the % of the signal for two intact alleles, calculated assuming a level of 0% per deleted (del) allele and 50% per wild type or undeleted flox allele. The intact level of 27.6 ± 1.7% (different from 50% at $P < 0.0001$ by 1 sample t-test) in the del/flox;Syn1-Cre mice indicates 45% deletion of the flox alleles. (B) *Adora1* RNA was measured in the same samples, assuming expression of 0% per del allele and 50% per wild type or undeleted flox allele. The RNA level of 32.6 ± 2.8% (different from 50% at $P < 0.0001$ by 1 sample t-test) in the del/flox;Syn1-Cre mice indicates 35% reduction per flox allele compared to wild type levels. DNA and RNA were extracted from one-half of the brain without cerebellum and brain stem and assayed as detailed in Methods. Data are mean ± SEM.

## Results

### No effect on baseline body temperature from neuronal loss of A$_1$AR

To distinguish between adenosine action on glia [27] and neurons, we produced mice carrying both floxed *Adora1* and Syn1-Cre to selectively ablate A$_1$AR function in neurons. The effect of neuronal deletion was studied in pooled *Adora1$^{fl/fl}$*;Syn1-Cre and *Adora1$^{del/fl}$*;Syn1-Cre mice (hereafter referred to collectively as *Adora1$^{fl}$*;Syn1-Cre, where del indicates a germline-deleted allele). In whole brain, Syn1-Cre produced 45% deletion per flox allele (Fig 1A). Concomitantly, Syn1-Cre reduced *Adora1* mRNA levels by 35% per flox allele (Fig 1B). Since Syn1-Cre is selectively expressed in differentiated neurons [31], these data are consistent with neuronal deletion of *Adora1$^{fl}$*, with the remaining expression likely derived from non-Syn1-Cre-expressing (non-neuronal) cells.

At baseline, there was no difference between control and *Adora1$^{fl}$*;Syn1-Cre littermate mice, matched for sex, in body temperature (Tb) during light or dark phase, or in Tb variability (as Tb span, the difference between the 95$^{th}$ and 5$^{th}$ Tb percentiles). There was also no effect of neuronal deletion on the level of physical activity or its diurnal rhythm (Table 1).

### A$_1$AR agonist-induced hypothermia is mediated by both peripheral and central neurons

Cl-ENBA is a selective A$_1$AR agonist. Peripherally-dosed Cl-ENBA (3 mg/kg, i.p.) caused robust hypothermia in *Adora1$^{fl}$* controls (-4.93 ± 0.17˚C vs vehicle at 0 to 60 minutes) and *Adora1$^{fl}$*;Syn1-Cre (-3.50 ± 0.53˚C vs vehicle) mice (Fig 2A–2C). The hypothermia in *Adora1$^{fl}$*; Syn1-Cre mice was attenuated compared to controls ($P = 0.02$). Since Cl-ENBA at high doses can also cause hypothermia via activation of mast cell A$_3$AR [21], we tested a lower Cl-ENBA dose (1 mg/kg; -3.33 ± 0.50˚C in controls vs. -1.83 ± 0.55˚C in *Adora1$^{fl}$*;Syn1-Cre mice; $P = 0.06$) (Fig 2D–2F). The similar level of partial loss of Cl-ENBA-induced hypothermia at

**Table 1. Deletion of neuronal A$_1$AR has no effect on baseline body temperature or physical activity.**

| Sex | *Adora1*$^{fl}$ Female | *Adora1*$^{fl}$;Syn1-Cre Female | P | *Adora1*$^{fl}$ Male | *Adora1*$^{fl}$;Syn1-Cre Male | P |
|---|---|---|---|---|---|---|
| N | 4 | 5 | | 7 | 8 | |
| Tb, light phase (˚C) | 36.17 ±0.07 | 36.40 ±0.16 | 0.27 | 35.67 ±0.16 | 35.76 ±0.12 | 0.64 |
| Tb, dark phase (˚C) | 37.37 ±0.17 | 37.39 ±0.16 | 0.94 | 36.77 ±0.13 | 36.93 ±0.05 | 0.25 |
| ΔTb, dark-light (˚C) | 1.21 ±0.10 | 0.99 ±0.25 | 0.50 | 1.10 ±0.10 | 1.17 ±0.11 | 0.64 |
| Tb span (˚C) | 2.58 ±0.15 | 2.09 ±0.45 | 0.38 | 2.28 ±0.08 | 2.49 ±0.14 | 0.23 |
| Activity, light phase (counts) | 6.8 ±0.5 | 8.1 ±0.6 | 0.18 | 5.4 ±0.3 | 6.4 ±0.4 | 0.09 |
| Activity, dark phase (counts) | 19.6 ±1.1 | 25.0 ±3.5 | 0.23 | 14.5 ±1.0 | 15.5 ±1.1 | 0.51 |

Data collected from 5-month old mice over a continuous 96-hour interval. Tb span is the 95[th] minus the 5[th] percentiles. Data are mean ± SEM. *P* values are from unpaired t-tests comparing genotype within sex.

the two doses suggests that i.p. Cl-ENBA is causing hypothermia partially through neuronal A$_1$AR.

SPA is a non-brain penetrant selective A$_1$AR agonist [37]. Peripherally dosed SPA (1 mg/kg, i.p.) caused hypothermia in wild-type mice that was lost in *Adora1*$^{-/-}$ mice (Fig 3A–3C). In *Adora1*$^{fl}$;Syn1-Cre mice, SPA-induced hypothermia was greatly diminished and not statistically different from vehicle treatment (Fig 3D–3F). These results suggest that hypothermia can be elicited via A$_1$AR expressed by neurons that are outside the blood-brain barrier.

We next tested the ability of a low i.c.v. dose of A$_1$AR agonist to produce hypothermia. Central administration of Cl-ENBA (3.06 μg/mouse; ~0.1 mg/kg) caused hypothermia in control but not *Adora1*$^{fl}$;Syn1-Cre mice (-2.30 ± 0.43˚C vs -0.23 ± 0.17˚C; *P* = 0.001) (Fig 4A–4C). This demonstrates that A$_1$AR agonists can cause hypothermia by acting directly on brain neurons.

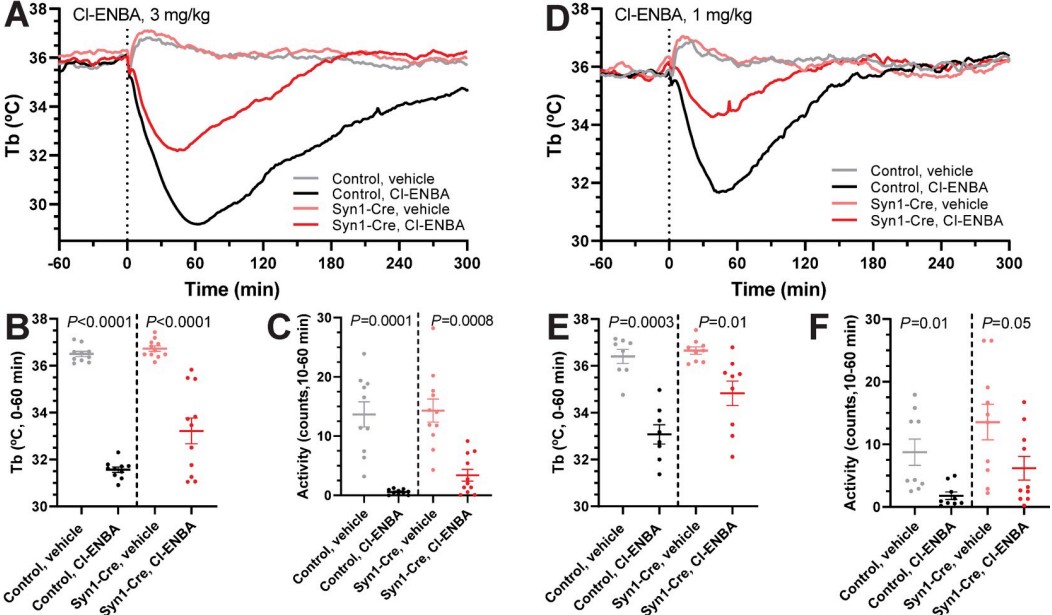

**Fig 2. Neuronal A$_1$AR partially mediate Cl-ENBA induced hypothermia.** (A-C) Core body temperature (Tb) and physical activity response to Cl-ENBA (3 mg/kg, i.p.) or vehicle in *Adora1*$^{fl}$ (n = 10) and *Adora1*$^{fl}$;Syn1-Cre (n = 11) mice. (D-F) Tb and physical activity response to Cl-ENBA (1 mg/kg, i.p.) or vehicle in *Adora1*$^{fl}$ (n = 8) and *Adora1*$^{fl}$;Syn1-Cre (n = 9) mice. Data are mean ± SEM (SEM is omitted for visual clarity in A and D). *P* values calculated by paired *t*-test comparing drug vs vehicle within mouse.

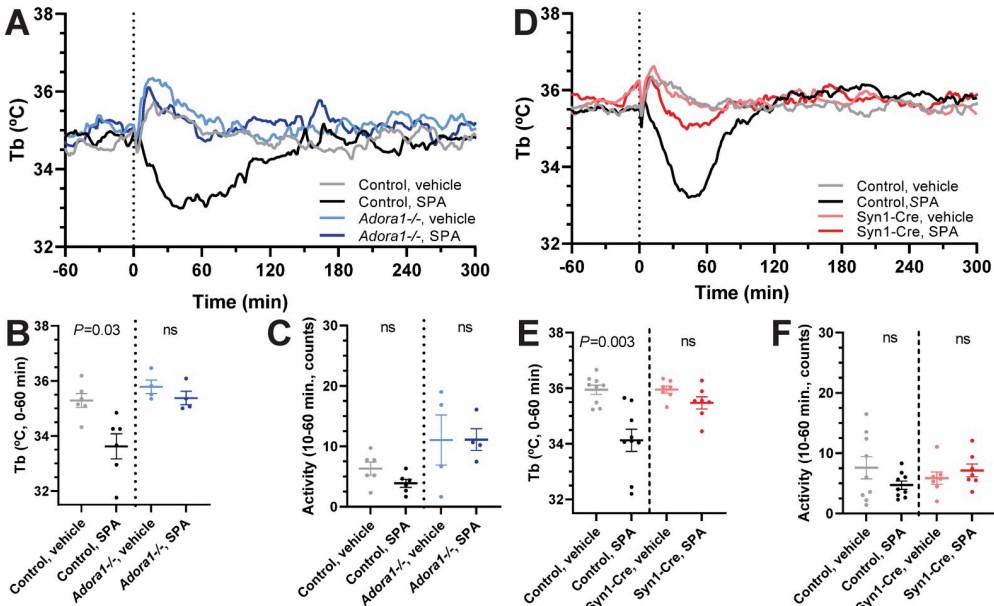

**Fig 3. SPA-induced hypothermia is lost in global *Adora1⁻/⁻* and attenuated in *Adora1ᶠˡ*;Syn1-Cre mice.** (A-C) Tb and physical activity response to SPA (1 mg/kg, i.p.) in control (n = 6) and *Adora1⁻/⁻* (n = 4) mice. (D-F) Tb and physical activity response to SPA (1 mg/kg, i.p.) in *Adora1ᶠˡ* (control, n = 9) and *Adora1ᶠˡ*;Syn1-Cre (Syn1-Cre, n = 7) mice. Data are mean ± SEM (SEM is omitted for visual clarity in A and D). *P* values were calculated by paired *t*-test comparing drug vs vehicle within mouse.

## Activation of Adora1-Cre-expressing neurons in the dorsomedial hypothalamus increased body temperature

Since A₁AR is widely distributed in the brain, we wished to probe specific regions for their potential role in driving A₁AR agonist-induced hypothermia. Adora1-Cre mice were produced by targeted insertion of a Cre recombinase gene into the endogenous *Adora1* locus, preserving the coding region (Methods, Fig 5A). The mice were bred with reporter mice expressing GFP in a Cre-dependent manner. The resulting GFP expression was widespread and particularly high in the cortex, hippocampus, and thalamus (Fig 5B). This GFP reporter expression pattern matches that reported for A₁AR ligand binding activity [26].

The preoptic area (POA) is a region of the brain with major control over Tb; other areas also contribute [38]. Local injection of A₁AR agonist has been shown to produce hypothermia in hypothalamic sites, including POA and dorsomedial hypothalamus (DMH) [16], so we focused on these regions. A₁AR is coupled to Gi/o, therefore we used chemogenetics to selectively target Adora1-Cre neurons with viral expression of a Gi-coupled designer receptor exclusively activated by designer drug (DREADD), hM4Di, to mimic the endogenous inhibitory A₁AR signaling. Mice with virally-delivered hM4Di in the POA were treated with DREADD agonist, clozapine-N-oxide (CNO), or vehicle. No effect of CNO on Tb was observed (Fig 6A–6C). All mice had at least unilateral virus expression in the medial preoptic area, and some mice had virus expression also in the medial septal nuclei and/or diagonal band of Broca. Thus, agonism of an inhibitory DREADD in POA^Adora1 neurons did not cause hypothermia.

We next tested activation of an excitatory Gq-coupled DREADD, hM3Dq, in Adora1-Cre-expressing cells. As endogenous A₁AR signaling is inhibitory, an increase in Tb due to chemogenetic activation of stimulatory Gq would be a concordant observation. Activation of POA^Adora1 neurons with CNO had no effect on Tb (Fig 7A–7C). All mice had unilateral or bilateral virus

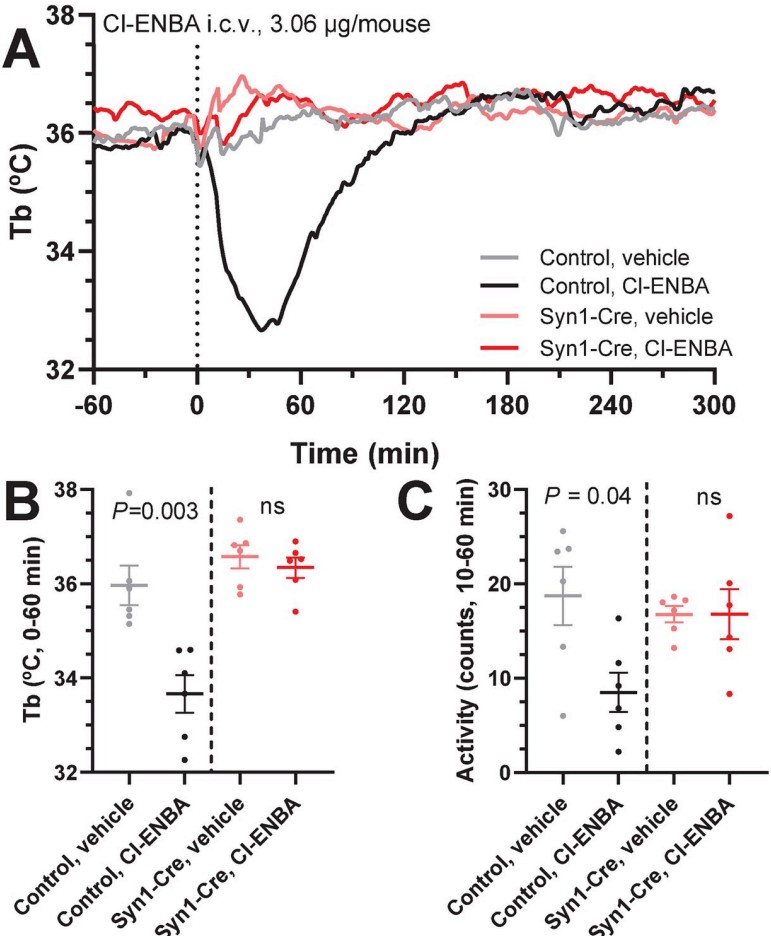

**Fig 4. Central administration of Cl-ENBA does not cause hypothermia in *Adora1^fl*;Syn1-Cre mice.** (A-C) Tb and physical activity response to Cl-ENBA (3.06 µg/mouse, i.c.v.) or vehicle in *Adora1^fl* (control, n = 6) and *Adora1^fl*; Syn1-Cre (Syn1-Cre, n = 6) mice. Data are mean ± SEM (SEM is omitted for visual clarity in A). *P* values were calculated by paired *t*-test comparing drug vs vehicle within mouse.

expression in the medial preoptic area, with virus expression in varying extent throughout the anterior hypothalamus. In contrast, activation of DMH^Adora1 neurons expressing hM3Dq with CNO increased Tb (Fig 7D–7F). All mice had unilateral or bilateral virus expression restricted to the DMH/dorsal hypothalamic area. Thus, the DMH is a potential candidate region for neurons driving A$_1$AR agonist-induced hypothermia.

## Discussion

The brain controls Tb. Thus, we had anticipated identifying an A$_1$AR-expressing neuron population, possibly in the POA, as the driver of A$_1$AR agonist-induced hypothermia. However, our results suggest that A$_1$AR agonism can cause hypothermia acting at neuronal A$_1$AR both inside and outside the brain, demonstrating that adenosine can cause hypothermia via more than one A$_1$AR neuronal mechanism.

### Central mechanisms of A$_1$AR hypothermia

While A$_1$AR is found in microglia, oligodendrocytes, astrocytes, and neurons [39], the loss-of-function experiments demonstrate that neurons (Syn1-Cre-positive cells) are required for

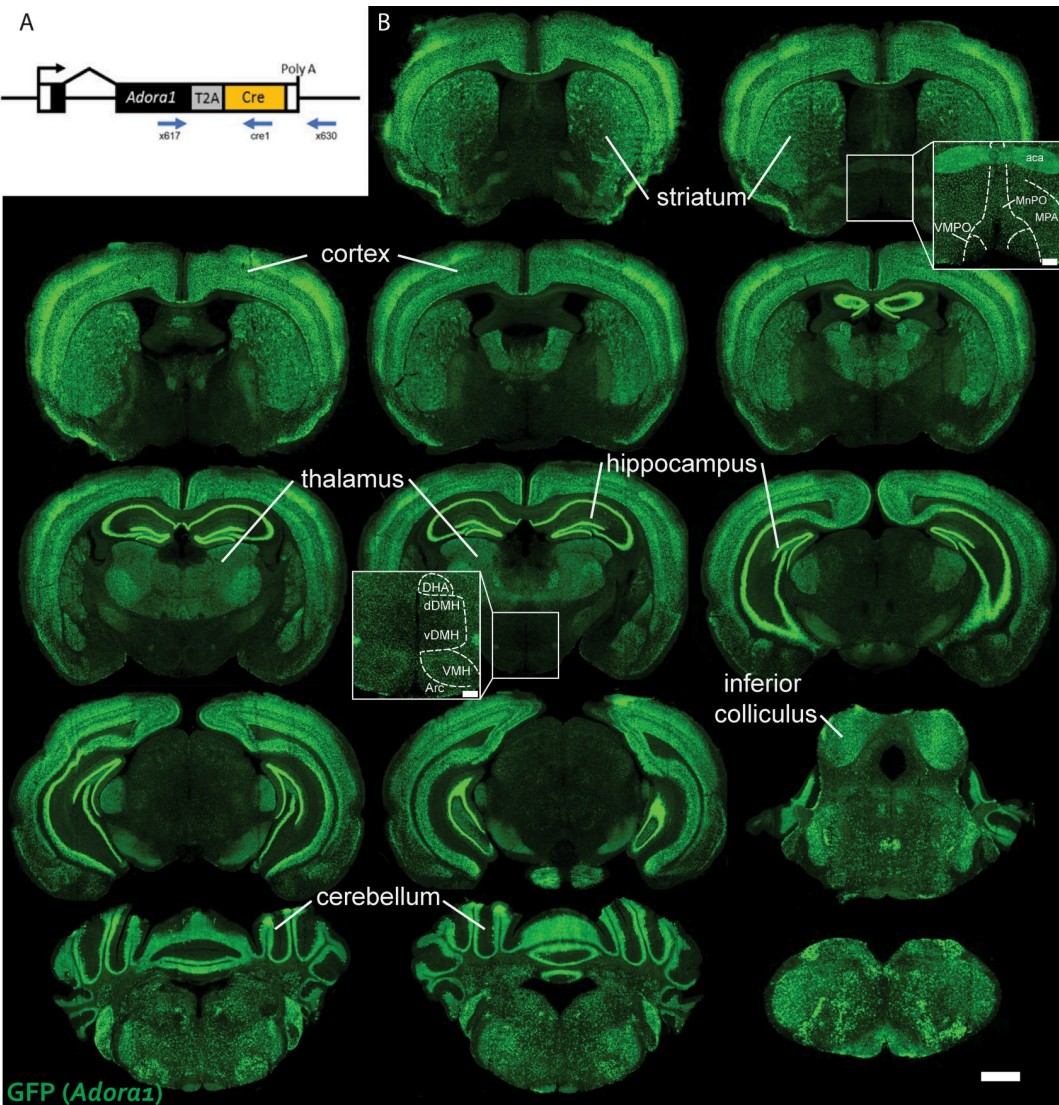

**Fig 5. Expression pattern of Adora1-Cre in mouse brain.** *(A)* Schematic representation of the Adora1-Cre allele, showing the Adora1coding region in black, T2A peptide in gray, and Cre recombinase in orange. Primers for PCR-based genotyping are also indicated. Not to scale. (B) Representative brain sections showing Cre-dependent GFP expression in Adora1-Cre;Ai6 mouse brain. Scale bar is 1 mm. Inset scale bar is 200 μm. MnPO, median preoptic nucleus; MPA, medial preoptic area; VMPO, ventromedial preoptic area; DHA, dorsal hypothalamic area; dDMH, dorsal part of dorsomedial hypothalamus; vDMH, ventral part of dorsomedial hypothalamus; VMH, ventromedial hypothalamus.

hypothermia caused by central administration of an A$_1$AR agonist. Activation of different defined populations of POA neurons can increase or decrease Tb [12, 40–44]. In addition, local A$_1$AR agonist administration to this region causes hypothermia [16]. Thus, we had expected that chemogenetic inhibition or activation of POA$^{Adora1}$ neurons would affect Tb, but this was not observed. The effect on Tb from stimulation of POA$^{Adora1}$ neurons may be more variable than in controls, suggesting that there may be multiple POA$^{Adora1}$ neuron populations with differing effects on Tb.

Chemogenetic activation of Adora1-Cre-expressing neurons of the DMH, another known thermoregulatory center, uniformly increased Tb, consistent with reports that disinhibition and activation of DMH neurons increase Tb [33, 40, 45]. One interpretation of our results is

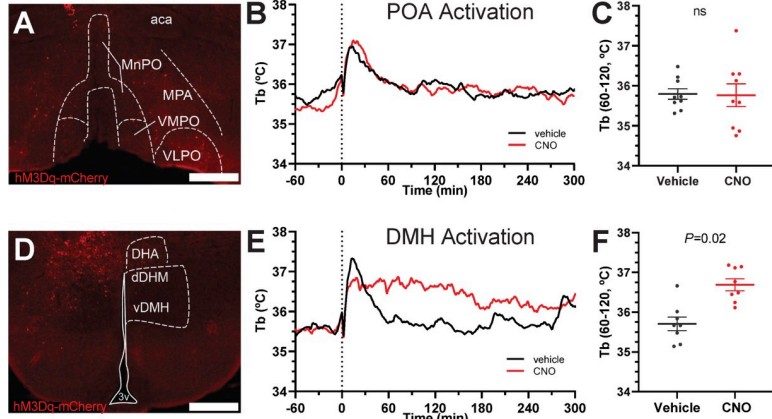

**Fig 6. Chemogenetic inhibition of POA$^{Adora1}$ neurons has no effect on Tb.** (A) Example of hM4Di-mCherry expression in Adora1-Cre mouse injected with AAV8-hSyn-DIO-hM4Di-mCherry in the POA. Scale bar is 500 μm. (B) Tb response to CNO (1 mg/kg, i.p.) vs. vehicle in Adora1-Cre mice with hM4Di-mCherry in POA (n = 7). (C) Mean Tb at 0 to 60 minutes after dosing. Data are mean ± SEM (SEM is omitted for visual clarity in B). *P* value was calculated by paired *t*-test comparing drug vs vehicle within mouse. Aca, anterior commissure; MnPO, median preoptic area; MPA, medial preoptic area; VMPO, ventromedial preoptic area.

that DMH$^{Adora1}$ neurons are driving A$_1$AR agonist-induced hypothermia, as activation of these neurons (opposing the inhibitory signaling from endogenous A$_1$AR activation) increased Tb. Another interpretation is that the chemogenetically manipulated neurons have a role in thermoregulation, but this is not directly related to pharmacologic A$_1$AR agonist-induced hypothermia.

Our results do not rule out the possibility of a single, specific nucleus mediating central A$_1$AR hypothermia. However, with widespread expression throughout the brain, we hypothesize that expression of A$_1$AR may not identify a discrete neuronal population that drives hypothermia. Rather, adenosine acting on A$_1$AR may be a general mechanism to reduce local neuronal activity, with the effect on Tb depending on the particular neuronal subpopulation's direct or indirect contributions to thermoregulation.

**Fig 7. Chemogenetic activation of DMH$^{Adora1}$, but not POA$^{Adora1}$ neurons increases Tb.** A-C tests activation of the POA. (A) Example of hM3Dq-mCherry expression in Adora1-Cre mouse injected with AAV8-hSyn-DIO-hM3Dq-mCherry in the POA. (B) Tb response to CNO (1 mg/kg, i.p.) vs. vehicle in Adora1-Cre mice with hM3Dq-mCherry in POA (n = 11). (C) Mean Tb at 60 to 120 minutes after dosing with CNO (1 mg/kg, i.p.) vs. vehicle. D-F tests activation of the DMH. (D) Example of hM3Dq-mCherry expression in Adora1-Cre mouse injected with AAV8-hSyn-DIO-hM3Dq-mCherry in the DMH. (E) Tb response to CNO (1 mg/kg, i.p.) vs. vehicle in Adora1-Cre mice with hM3Dq-mCherry in DMH (n = 8). (F) Mean Tb at 60 to 120 minutes after dosing with CNO (1 mg/kg, i.p.) vs. vehicle (n = 8). Data are mean ± SEM (SEM is omitted for visual clarity in B and E). *P* values were calculated by paired *t*-test comparing drug vs vehicle within mouse. Scale bar is 500 μm. aca, anterior commissure; dDMH, dorsomedial hypothalamus, dorsal part; DHA, dorsal hypothalamic area; MnPO, median preoptic area; MPA, medial preoptic area; vDMH, dorsomedial hypothalamus, ventral part; VMPO, ventromedial preoptic area; VLPO, ventrolateral preoptic area.

## Peripheral mechanisms of A$_1$AR hypothermia

Our results indicate A$_1$AR agonist-induced hypothermia can be caused by neuronal A$_1$AR outside the central nervous system. The hypothermic effect of the non-brain penetrant, peripherally-restricted A$_1$AR agonist SPA was lost in *Adora1$^{fl}$*;Syn1-Cre mice, indicating that peripheral A$_1$AR agonist-induced hypothermia is mediated by cells expressing synapsin-1 (i.e. peripheral nerves). A$_1$AR is found throughout the peripheral nervous system (PNS): in motor and sensory nerve terminals of rats, as well as sympathetic and parasympathetic nerves [46–49]. Like central A$_1$AR, PNS A$_1$AR play a role in regulating numerous physiological processes (nociception, vascular tone, etc.) by providing an inhibitory tone on neurotransmission. Therefore, the effect of PNS A$_1$AR agonism on Tb depends on the function of the peripheral neuron being inhibited.

A$_1$AR is present in many non-neuronal cells (smooth muscle, epithelial, immune cells) and tissues (heart, adipose tissue, pancreas, kidney) that have a very wide variety of functions (reviewed in [1]). Agonism of non-neural A$_1$AR can produce physiologic effects such as hypotension that could secondarily decrease Tb. The lack of significant hypothermia produced by the peripheral A$_1$AR agonist SPA in *Adora1$^{fl}$*;Syn1-Cre mice suggests that while non-neuronal cells may contribute, neuronal A$_1$AR are required for hypothermia by this mechanism. Further research will be needed to identify the specific neuronal populations involved.

## Adenosine as a physiologic and general danger signal

Adenosine can act at each of the four adenosine receptors to cause hypothermia. Activation of A$_3$AR on peripheral mast cells causes hypothermia via histamine release, followed by activation of H$_1$ receptors [22, 50, 51]. Agonism at peripheral A$_{2A}$AR or central A$_{2B}$AR both cause hypothermia [20, 52]. However, the specific cells that elicit hypothermia when directly activated by A$_{2A}$AR and A$_{2B}$AR agonists have not been identified. Including A$_1$AR on central and on peripheral neurons, there are five identified paths by which adenosine agonism elicits hypothermia. When hypothermia is studied in sufficient detail, it is accompanied by hypometabolism, decreased physical activity, and hypotension. While hypometabolism is likely required to achieve hypothermia, different driving physiology (eg., hypotension vs torpor) can produce hypothermia.

Adenosine is a ubiquitous molecule that acts locally to signal metabolic stress or injury. The distinct mechanism(s) by which activation of each AR causes hypothermia suggests that adenosine signaling is harnessed to minimize damage, regardless if the signaling occurs centrally or peripherally. Hypothermia can be due to direct adenosine agonism on thermoregulatory neurons, or a secondary effect of other adenosine-driven physiology. Taken with previous studies, these results demonstrate that at least five distinct mechanisms of hypothermia mediated by adenosine receptors exist. The redundancy in AR signaling highlights the importance of hypothermia as a component of protective response mechanisms.

## Supporting information

**S1 Data.**
(XLSX)

## Acknowledgments

We thank Alice Franks, Naili Liu, Yuning Huang, and Zhenzhong Cui for experimental support, and Chengyu Liu of the NHLBI Transgenic Core for collaborating in generating the Adora1-Cre mouse.

## Author Contributions

**Conceptualization:** Marc L. Reitman.

**Investigation:** Haley S. Province, Cuiying Xiao, Allison S. Mogul, Ankita Sahoo, Ramón A. Piñol, Oksana Gavrilova.

**Visualization:** Marc L. Reitman.

**Writing – original draft:** Haley S. Province, Cuiying Xiao, Marc L. Reitman.

**Writing – review & editing:** Haley S. Province, Cuiying Xiao, Allison S. Mogul, Ankita Sahoo, Kenneth A. Jacobson, Ramón A. Piñol, Oksana Gavrilova, Marc L. Reitman.

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
