## [Decision Letter · Decision Letter 0]

28 Oct 2020

PONE-D-20-31332

Activation of neuronal adenosine A1 receptors causes hypothermia through central and peripheral mechanisms

PLOS ONE

Dear Dr. Reitman,

Thank you for submitting your manuscript to PLOS ONE. After careful consideration, I feel that this submission has merit but does not fully meet PLOS ONE’s publication criteria as it currently stands. Therefore, I wouwld invite you to submit a revised version of the manuscript that addresses the points raised during the review process.

Please see the attached comments from review below. In particular, I would like you to improve the introduction for clarity and to address the concerns about the missing positive control for the experiment shown in Figure 6.

We look forward to receiving your revised manuscript.

Kind regards,

Edward E Schmidt

Academic Editor

PLOS ONE

Journal Requirements:

2. To comply with PLOS ONE submissions requirements, please provide methods of sacrifice in the Methods section of your manuscript.

Reviewers' comments:

Reviewer's Responses to Questions

**Comments to the Author**

1. Is the manuscript technically sound, and do the data support the conclusions?

Reviewer #1: Yes

2. Has the statistical analysis been performed appropriately and rigorously? 

Reviewer #1: Yes

3. Have the authors made all data underlying the findings in their manuscript fully available?

Reviewer #1: Yes

4. Is the manuscript presented in an intelligible fashion and written in standard English?

Reviewer #1: Yes

5. Review Comments to the Author

Reviewer #1: Major points: This article explores the relationship between an adenosine receptor (A1AR), regions of the brain and hypothermia. Overall, the article would be much stronger and would benefit from more in-depth explanation on what the target of the study is. The last line of the introduction (line 75) indicates that the study will be looking at where A1AR agonist will act to cause hypothermia, and this is assuming the authors mean where in the brain, but there is not enough introduction about what is known currently about A1AR/hypothermia/brain localization in the intro. Where is A1AR localized in the brain? What regions are believed to control hypothermia and which have not been explored? As the authors noted, other studies have also “suggested a role in hypothermia for A1AR in the preoptic area and other brain regions”, so what is unique about this study?

Intro - It should be clarified that the Adora1 locust encodes the protein of interest, A1AR.

Figure 1 – Significance of the data should be noted with p values

Line 239 – “Since A1AR are widely” change are to “is”

Figure 3, a new abbreviation (A1KO) is used, but this is not discussed in the rest of the paper or defined. Authors should add definition or use Adora-/- or Adorafl as in the rest of the paper.

Line 229 – consider rephrasing “A1AR agonist-induced hypothermia is also caused by acting on brain neurons.” It is unclear in the currently written form

Figure 5 – This figure would benefit from labeling of different brain regions, as in the other figures, especially the noted cortex, hippocampus, and thalamus

Figure 6 – A positive control demonstrating that the AAV8-hSyn-DIO-hM4Di-mCherry does cause hypothermia in other areas of the brain would be beneficial. Alternatively, some evidence that the hM4Di inhibitor is actually being expressed in addition to the mCherry would be beneficial. It is impossible to know if the hM4Di didn’t cause hypothermia in the POA or if the hM4Di segment is inactive. Also, some explanation of the uneven distribution of mCherry in Figure 6A would be beneficial. Is the VMPO special for some reason that it has a lot of AAV8-hSyn-DIO-hM4Di-mCherry present?

Figure 7 – Comparison of Fig7B and 7E would be easier with a clear title or label of the variable that is changed (testing activation in the POA vs DMH).

Lines 276 and 279 – the wording “at least unilateral” should be changed to just “unilateral”, there’s nothing more than unilateral.

Line 280 – The authors should clarify that while they induced hyperthermia, it is not necessarily given that the same region and neuron subpopulation would control hypothermia.

Figure 7B – Fix “CNO” in legend

6. PLOS authors have the option to publish the peer review history of their article (what does this mean?). If published, this will include your full peer review and any attached files.

Reviewer #1: No

---

## [Editor Report · Decision Letter 1]

30 Nov 2020

PONE-D-20-31332R1

Activation of neuronal adenosine A1 receptors causes hypothermia through central and peripheral mechanisms

PLOS ONE

Dear Dr. Reitman,

Thank you for submitting your manuscript to PLOS ONE. After careful review of your resubmission, I am satisfied with your responses to the review, but I see that you have not included a statement in the text about public availability of the mouse models used in this study. I presume that the null and floxed alleles of Adora1 that you used can be obtained by contacting Drs. Schnermann and Greene, respectively, via the information given on the manuscripts you have cited for these models. Can you please revise the manuscript, however, to instruct readers how they can obtain the ADAR1Cre mouse line that you developed in this study?  Thank you!

A rebuttal letter that responds to each point raised by the academic editor and reviewer(s) my request. You should upload this letter as a separate file labeled 'Response to Reviewers' Editor's request.A marked-up copy of your manuscript that highlights changes made to the original version. You should upload this as a separate file labeled 'Revised Manuscript with Track Changes'.An unmarked version of your revised paper without tracked changes. You should upload this as a separate file labeled 'Manuscript'.

If applicable, I recommend that you deposit your laboratory protocols in protocols.io to enhance the reproducibility of your results. Protocols.io assigns your protocol its own identifier (DOI) so that it can be cited independently in the future. For instructions see: http://journals.plos.org/plosone/s/submission-guidelines#loc-laboratory-protocols

I look forward to receiving your revised manuscript.

Kind regards,

Edward E Schmidt

Academic Editor

PLOS ONE

---

## [Author Response · Author response to Decision Letter 1]

30 Nov 2020

From the Editor:

“I see that you have not included a statement in the text about public availability of the mouse models used in this study. I presume that the null and floxed alleles of Adora1 that you used can be obtained by contacting Drs. Schnermann and Greene, respectively, via the information given on the manuscripts you have cited for these models. Can you please revise the manuscript, however, to instruct readers how they can obtain the ADAR1Cre mouse line that you developed in this study?”

Indeed, the null and floxed alleles of Adora1 can be obtained by contacting Drs. Schnermann and Greene, respectively.

We have added a sentence (line 104) “The Adora1-Cre mouse is available from the corresponding author.” With acceptance of the manuscript, we will deposit the mouse in a repository.

---

## [Editor Report · Decision Letter 2]

2 Dec 2020

Activation of neuronal adenosine A1 receptors causes hypothermia through central and peripheral mechanisms

PONE-D-20-31332R2

Dear Dr. Reitman,

Thank you for your rapid revision in response to my request.  I am pleased to inform you that your manuscript has been judged scientifically suitable for publication and will be formally accepted for publication once it meets all outstanding technical requirements.

Kind regards,

Edward E Schmidt

Academic Editor

PLOS ONE
---

## [Editor Report · Acceptance letter]

7 Dec 2020

PONE-D-20-31332R2 

Activation of neuronal adenosine A1 receptors causes hypothermia through central and peripheral mechanisms 

Dear Dr. Reitman:

I'm pleased to inform you that your manuscript has been deemed suitable for publication in PLOS ONE. Congratulations! Your manuscript is now with our production department. 

Kind regards, 

on behalf of

Dr. Edward E Schmidt 

Academic Editor

PLOS ONE